# The Therapeutic Potential of Apigenin

**DOI:** 10.3390/ijms20061305

**Published:** 2019-03-15

**Authors:** Bahare Salehi, Alessandro Venditti, Mehdi Sharifi-Rad, Dorota Kręgiel, Javad Sharifi-Rad, Alessandra Durazzo, Massimo Lucarini, Antonello Santini, Eliana B. Souto, Ettore Novellino, Hubert Antolak, Elena Azzini, William N. Setzer, Natália Martins

**Affiliations:** 1Student Research Committee, School of Medicine, Bam University of Medical Sciences, Bam 44340847, Iran; bahar.salehi007@gmail.com; 2Dipartimento di Chimica, “Sapienza” Università di Roma, Piazzale Aldo Moro 5, 00185 Rome, Italy; alessandro.venditti@gmail.com; 3Department of Medical Parasitology, Zabol University of Medical Sciences, Zabol 61663-335, Iran; 4Institute of Fermentation Technology and Microbiology, Lodz University of Technology, Wolczanska 171/173, 90-924 Lodz, Poland; dorota.kregiel@p.lodz.pl; 5Food Safety Research Center (salt), Semnan University of Medical Sciences, Semnan 35198-99951, Iran; 6CREA-Research Centre for Food and Nutrition, Via Ardeatina 546, 00178 Rome, Italy; 7Department of Pharmacy, University of Napoli Federico II, Via D. Montesano 49, 80131 Napoli, Italy; ettore.novellino@unina.it; 8Faculty of Pharmacy of University of Coimbra Azinhaga de Santa Comba, Polo III-Saúde 3000-548 Coimbra, Portugal; ebsouto@ebsouto.pt; 9CEB-Centre of Biological Engineering, University of Minho, Campus de Gualtar, 4710-057 Braga, Portugal; 10Department of Chemistry, University of Alabama in Huntsville, Huntsville, AL 35899, USA; setzerw@uah.edu; 11Faculty of Medicine, University of Porto, Alameda Prof. Hernâni Monteiro, 4200-319 Porto, Portugal; ncmartins@med.up.pt; 12Institute for Research and Innovation in Health (i3S), University of Porto, 4200-135 Porto, Portugal

**Keywords:** apigenin, flavonoids, chronic diseases, diabetes, cancer

## Abstract

Several plant bioactive compounds have exhibited functional activities that suggest they could play a remarkable role in preventing a wide range of chronic diseases. The largest group of naturally-occurring polyphenols are the flavonoids, including apigenin. The present work is an updated overview of apigenin, focusing on its health-promoting effects/therapeutic functions and, in particular, results of in vivo research. In addition to an introduction to its chemistry, nutraceutical features have also been described. The main key findings from in vivo research, including animal models and human studies, are summarized. The beneficial indications are reported and discussed in detail, including effects in diabetes, amnesia and Alzheimer’s disease, depression and insomnia, cancer, etc. Finally, data on flavonoids from the main public databases are gathered to highlight the apigenin’s key role in dietary assessment and in the evaluation of a formulated diet, to determine exposure and to investigate its health effects in vivo.

## 1. Introduction

Chronic diseases, such as cancer, stroke, diabetes, Alzheimer’s disease, depression, and age-related function decline, are major public health burdens worldwide, especially in developed countries. It is believed that a combination of regular physical activity and healthy diet can prevent these various diseases, as well as help in fighting already existing diseases. Particular attention has been paid to a diet based on fruits and vegetables, which are sources of natural bioactive compounds with pro-health properties. The largest group of naturally-occurring polyphenols are the flavonoids, which include flavones, flavonols, flavanones, flavanols, isoflavonoids, and anthocyanidins [1,2,3,4]. Flavonoids are characterized by broad biological activities, demonstrated in numerous mammalian systems in vitro and in vivo. These compounds act as free-radical scavengers and antioxidants, exhibiting anti-mutagenic, anti-inflammatory, and antiviral effects [5,6,7,8,9,10,11]. What is more, flavonoids are able to reduce plasma levels of low-density lipoproteins, inhibit platelet aggregation, and reduce cell proliferation. These properties result, inter alia, from their mechanisms of action: inhibiting the cell cycle, diminishing oxidative stress, improving detoxification enzymes, inducing apoptosis, and stimulating the immune system. Of all the flavonoids, apigenin (4′,5,7-trihydroxyflavone) is one of the most widely distributed in the plant kingdom, and one of the most studied phenolics. Apigenin is present principally as glycosylated in significant amount in vegetables (parsley, celery, onions) fruits (oranges), herbs (chamomile, thyme, oregano, basil), and plant-based beverages (tea, beer, and wine) [12]. The present review is focused on the health-promoting effects of apigenin, in particular through in vivo research.

## 2. Apigenin Chemistry

Flavonoids comprise a class of naturally-occurring phytochemicals in almost all plants tissues, where they play different functions. For example, they protect plants from harmful sunlight radiation, defend against pathogens and herbivory, regulate plant metabolism, and serve as visual attractors for pollinators. Currently, more than 6000 different compounds belonging to the flavonoids have been described [13]. Chemically, they can also be classified as polyphenols since they very often have one or more hydroxyl substituents in their structure. They are composed of a flavane nucleus of 15 carbon atoms (C6-C3-C6) and are diphenyl-propanoids (Figure 1). The C6 and C3 moieties are arranged to form two fused rings in which the first is an oxygen-containing heterocycle and the second one is a benzene ring constituting a phenylchromane nucleus (2,3-dihydro-2-phenylchroman-4-one). To the base skeleton of phenylchromane a second phenyl substituent is linked and, according to the bond position (C2, C3, C4), flavanes, isoflavanes, and neoflavanes (Figure 1), respectively, may be obtained. On the other hand, on the basis of the substitution patterns (i.e., oxygenation) of the three rings, several sub-classes of flavonoids can be formed (flavones, flavonols, flavanones, flavanonols, flavans, flavan-3-ols, anthocyanidins, etc.) (Figure 1). Flavonoids may exist as both aglycones and prenylated and methyl ethers, and in glycosylated forms with sugar residues that can be linked at several positions of the three rings and form both *O*- and *C*-glycosides [14]. Apigenin (4′,5,7-trihydroxyflavone) (Figure 2) is one of the most widespread flavonoids in plants and formally belongs to the flavone sub-class. Plants belonging to the *Asteraceae*, such as those belonging to *Artemisia* [15], *Achillea* [16,17], *Matricaria* [18], and *Tanacetum* [19] genera, are the main sources of this compound. However, species belonging to other families, such as the *Lamiaceae*, for instance, *Sideritis* [20] and *Teucrium* [21,22], or species from the *Fabaceae*, such as *Genista* [23], showed the presence of apigenin in the aglycone form and/or its *C*- and *O*-glucosides, glucuronides, *O*-methyl ethers, and acetylated derivatives (Figure 2). In some cases the chemotaxonomic relevance has also been demonstrated. In gymnosperms, apigenin derivatives are mostly present in dimeric forms, with apigenin residues variously coupled, e.g., with C‒C linkage as in cupressuflavone and amentoflavone (I-8, II-8″ and I-3′, II-8″, respectively), or C‒O linkage (I-4′, II-6″) as in hinokiflavone (Figure 2) [24,25,26,27]. Biogenetically, apigenin is a product of the phenylpropanoid pathway and may be obtained from both phenylalanine and tyrosine, two shikimate-derived precursors. From phenylalanine, cinnamic acid is formed by non-oxidative deamination and then by oxidation at C-4, which is then converted to *p*-coumaric acid, whereas from tyrosine *p*-coumaric acid is formed directly by deamination. After activation with CoA, *p*-coumarate is condensed with three malonyl-CoA residues and then subjected to aromatization by chalcone synthase to form chalcone, which is further isomerized by chalcone isomerase to form naringenin; finally, a flavanone synthase oxidizes naringenin to apigenin (Figure 3) [28,29,30,31,32,33]. Flavonoids, in general, are widely known for their antioxidant properties and a huge number of reports in the literature have reported the antioxidant properties of apigenin [34]. In addition, anti-hyperglycemic [35], anti-inflammatory [36], and (in myocardial ischemia) anti-apoptotic effects [37] have been reported. A recent review has summarized several of its biological effects such as cytostatic and cytotoxic activities toward various cancer cells, anti-atherogenic and protective effects in hypertension, cardiac hypertrophy, autoimmune myocarditis, among others [38]. The interest in the various biological activities of apigenin is rising, and it has led to the development of efficient methods of extraction from its natural sources, also with the use of modern extractive approaches, for instance dynamic maceration process [39] and ionic liquid analogs (deep eutectic solvents) as unconventional extractive solvents [40].

## 3. Apigenin Biological Potential: from Molecular Aspects to Health Promoting Targets

### 3.1. Apigenin as a Nutraceutical: Introduction of the Concept and its Absorption, Distribution, Metabolism, and Excretion (ADME) Behavior

Among the wide variety of phenolic compounds, apigenin is one of the most renowned, with countless nutritional and organoleptic characteristics. Nonetheless and more interestingly, it can also contribute with its beneficial health properties, which could lead to a possible inclusion in nutraceutical formulations [12,41]. Due to apigenin’s variety of pharmacological activities and importance for human health, a deepen knowledge of its mechanism of action would be of utmost importance for possible nutraceutical applications. The term nutraceutical, originally coined by Stephen De Felice [42], is, in this context, intended as: (i) the phytocomplex for food or part of food of vegetal origin; and (ii) the pool of secondary metabolites for food or part of food of animal origin [43]. This novel concept’ definition has been proposed to better evaluate the standard term of the word nutraceutical, and to highlight the difference between nutraceuticals, food supplements, and the many other plant-food-derived compounds that claim health-promoting effects. Nutraceuticals must be administered in a proper pharmaceutical way to guarantee high bioavailability and efficacy, and used in areas that go “beyond the diet and before the drugs” [44]. Nutraceuticals form a growing and powerful toolbox that is triggering a revolution in the area of disease prevention and also in the treatment for some clinical situations, in particular for individuals who may not yet be eligible for conventional pharmaceutical therapy, e.g., with conditions linked to metabolic syndrome [43,44,45,46,47]. It is therefore necessary to unequivocally determine the definition of nutraceuticals, and to have an internationally shared regulation framework. It would also be advisable to determine nutraceuticals’ safety, modes of action and efficacy with clinical data before naming a product as “nutraceutical”, a term that must be substantiated by safety, no side effects, and proven health beneficial properties [48,49].

Till now, little evidence reports apigenin adverse metabolic reactions; consequently, its consumption through the diet is recommended. After ingestion, in order to exert its healing properties, a bioactive compound like apigenin undergoes several metabolic pathways and its pharmacokinetic behavior affects its tissue distribution and bioactivity. In nature, apigenin also occurs linked through C‒C or C‒O‒C bonds of dimeric forms. Between flavonoid aglycones and their glycosides there are different pharmacokinetic behaviors and healing outcomes. The effect of the *O*-glycosylation or *C*-glycosylation of apigenin can affect its metabolism in different ways and therefore affect its antioxidant potential and biological benefits. Cai et al. [50] reported a decrease in its antioxidant potential in vitro assays due to *O*-glycosylation of apigenin. Regarding the bioavailability of the apigenin-*C*-glycosides, Angelino et al. [51], reported the unchanged absorption of vitexin-2-*O*-xyloside (VOX), an apigenin-8-*C*-glucoside in a rat model. The apigenin-8-*C*-glycoside undergoes enterohepatic recirculation in addition to hydrolysis to the monoglycoside, reduction, and conjugation to form a bioavailable glucuronide.

A high number of studies carried out over the years have indicated that apigenin has many interesting pharmacological activities and nutraceutical potential. As an example, its properties as an antioxidant are well known, and it can also be a therapeutic agent to overcome diseases like inflammation, autoimmune, neurodegenerative disease, and even several types of cancers. It has a low intrinsic toxicity on normal versus cancerous cells, compared with other structurally related flavonoids [52,53]. Notwithstanding its importance, there is a lack of research related to beneficial health potential of apigenin for humans with respect, for example, to inflammation or cognitive performance, another relevant potential application of this substance. This is probably due to the fact that, despite the numerous positive effects, apigenin has a very low solubility in water (1.35 μg/mL) and high permeability [54]. This may limit the use of apigenin in in vivo studies. Several approaches to improve its solubility, including different delivery systems (liposomes, polymeric micelles, nanosuspension, and so on) [55,56,57,58], showed how solid dispersion for enhancing the solubility and dissolution of drugs with poor water solubility improved stability as well as dosing [58]; in particular, different injectable nanosized drug delivery systems have been developed, suggesting that nanocapsules may be a good approach to prolong apigenin’ pharmacological activity [59]. On the other hand, Azzini et al. [60] highlighted the low bioavailability and high metabolic transformation of some food components as an unsolved issue on the way to demonstrate a clear structure‒function relationship in regulating cellular physiology.

### 3.2. Healing Properties of Apigenin: An Overview

In recent years interest in apigenin as a beneficial and health-promoting agent has grown.

Recently Kashyap et al. [61] summarized the multiple therapeutic functions of apigenin by in vitro and in vivo systems. The different mechanisms underlying the potential therapeutic action of apigenin were explored, including cell cycle arrest, apoptosis, anti-inflammatory, and antioxidant function. Apigenin induces cell cycle arrest at different proliferation stages including G1/S-phase or G2/M phase by modulating the expression of different CDKs and other genes [62,63,64]. It is known that apigenin can regulate intrinsic apoptotic pathways, changing mitochondrial membrane potential and causing the release of cytochrome C in the cytoplasm, which subsequently forms APFA, activates caspase 3, and turns on apoptosis [65]. Otherwise, apigenin regulated extrinsic apoptotic pathways by involving caspase-8 activation. In cancer cells, apigenin activates apoptosis by modulating Bcl-2, Bax, STAT-3 and Akt proteins expression [66,67]. Apigenin promotes different anti-inflammatory pathways, including p38/MAPK and PI3K/Akt, as well as prevent the IKB degradation and nuclear translocation of the NF-κB, and reduce COX-2 activity [68,69,70]. In human cell cultures, apigenin has demonstrated the ability to inactivate nuclear factor kappa-light-chain-enhancer or to activate B cells (NF-κB), mediated by suppression of LPS-induced phosphorylation of the p65 subunit [71]. It is believed that apigenin decreases the expression of adhesion molecules, which is a defensive strategy against oxidative stress, such as free-radical scavenging [72]. Apigenin enhances the expression of anti-oxidant enzymes such as GSH-synthase, catalase, and SOD to counteract cellular oxidative and electrophilic stress. It also enhances the expression of phase II enzyme encoding genes by blocking the NADPH oxidase complex and their downstream target inflammatory genes and by increasing the expression of nuclear translocation of Nrf-2 [73,74,75]. Apigenin is also reported to induce the inhibition of metastasis and angiogenesis by interacting with the signaling molecules in the three major mitogen-activated protein kinase (MAPK) pathways: extracellular-signal-regulated kinase (ERK), c-Jun N-terminal kinases (JNK), and p38 in human cell culture models [76]. It is known that apigenin strongly decreased levels of interleukin 6 (IL-6), which in general acts as both a pro-inflammatory cytokine and an anti-inflammatory myokine, in lipopolysaccharide (LPS)-activated mouse macrophages. This flavonoid is also known for suppressing cluster of differentiation 40 (CD40), tumor necrosis factor (TNF-α), and IL-6 production via inhibition of interferon gamma (IFN-γ)-induced phosphorylation of signal transducers and activators of transcription 1 (STAT1) in murine microglia [77].

Furthermore, after absorption into the digestive tract, apigenin is able to reach the brain through the circulatory system, where it can cross the blood‒brain barrier before exerting its affinity with the GABA_A_-receptor and acting on the CNS, despite its action at the level of improving the clinical use of benzodiazepines is not clear [78,79,80]. Sloley et al. [81] reported the in vitro inhibition of rat-brain monoamine oxidases (MAOs) by apigenin, a family of enzymes of flavin-containing amine oxidoreductases, present in human neurons and astroglia. Unregulated MAO activity could be responsible for a number of psychiatric and neurological disorders and their inhibitors, like apigenin, work as antidepressant and antianxiety agents, as well as to treat Alzheimer’s and Parkinson’s disease.

Even though there has been a relatively large number of in vitro studies on apigenin properties, the number of in vivo studies using the mouse, rat, or hamster as a model is relatively small. The situation is even more unfavorable for clinical trials involving people. The number of such studies is extremely small, in particular in the case of the effect of this compound on cancer, which may be due to, among other factors, the ethical aspects. We summarized the main results reviewed in separate figures for animal (Figure 4A–C) and human (Figure 5) studies, respectively. 

#### 3.2.1. Antidiabetic Properties of Apigenin

The anti-diabetic properties of apigenin may be attributed to its capacity to inhibit α-glucosidase activity, increase secretion of insulin [107], to interact with and neutralize reactive oxygen species (ROS) in the cell [108], which together contribute to the prevention of diabetic complications [109]. Apigenin has also shown the ability to supply moderate nitric oxide (NO) to endothelial cells, thereby limiting the risk of endothelial cell injury and dysfunction from hyperglycemia [109].

Panda and Kar [110] confirmed the capacity of apigenin in regulating hyperglycemia, thyroid dysfunction, and lipid peroxidation in a diabetic animal model [110]. The administration of apigenin to alloxan-treated mice also suggested the hepatoprotective role of this nutraceutical compound, attributed to its capacity to increase the activity of cellular antioxidants, such as catalase (CAT) and superoxide dismutase (SOD), and glutathione (GSH). Similar results have been reported by Ren et al. [84], who demonstrated decreased levels of blood glucose, serum lipid, malonaldehyde, intercellular adhesion molecule-1, and insulin resistance index, increased SOD activity, and improved impaired glucose tolerance of apigenin when compared to a diabetic control group.

Panda and Kar [110] also showed that alloxan-induced elevation in serum cholesterol was also reverted by the administration of apigenin, while Ren et al. [84] reported that pathological damage in the thoracic aorta of the apigenin group was more remissive than the diabetic control group [84].

To demonstrate the cardioprotective effects of apigenin, the pathologic changes shown in diabetic cardiomyopathy-induced mice (i.e., enhanced cardiac dysfunction, fibrosis, overaccumulation of 4-hydroxynonenal followed by downregulation of Bcl2, GPx, and SOD, upregulation of MDA, cleaved caspase3, and pro-apoptotic protein Bax, and contribution to the translocation of NF-kappaB) could be reversed by treatment with apigenin in vivo [111]. Apigenin (20 mg/kg) administered to male albino Wistar rats improved renal dysfunction, oxidative stress, and fibrosis (decreased transforming growth factor-beta1, fibronectin, and type IV collagen) [82].

In another study, apigenin treatment prevented the hemodynamic variations, restored the left ventricular function, and reinstated a balanced redox status in vivo [112]. Rats were protected against myocardial injury by attenuating myonecrosis, edema, cell death, and oxidative stress.

Cazarolli et al. [83] studied the effect of apigenin-6-*C*-(2″-*O*-α-L-rhamnopyranosyl)- β-L-fucopyranoside, obtained from *Averrhoa carambola* L. leaves, on ^14^C-glucose uptake [83]. The authors reported in diabetic rats the acute effect of this compound on lowering blood glucose and stimulated glucose-induced insulin secretion after oral treatment in hyperglycemic rats.

#### 3.2.2. Apigenin’s Beneficial Role in Amnesia and Alzheimer’s Disease

Several natural bioactive compounds for improving learning and memory, as well as some active and passive anti-amyloid-β and anti-tau immunotherapies using synthetic peptides or monoclonal antibodies (mAb), have been reported as promising candidates for further treatment of patients with Alzheimer’s disease [113,114,115,116].

The recent review of Nabavi et al. [116] discussed the evidence from the various animal models and human clinical trials on the therapeutic potential of apigenin, in particular its antioxidant activity and potential role as a neuroprotective agent, as also its chemistry, pharmacokinetics, and metabolism in the context of depression, Alzheimer’s disease, and Parkinson’s disease [116].

Apigenin may induce muscle relaxation and sedation depending on the dose [117], and it is also active as an antioxidant, anti-inflammatory, anti-amyloidogenic, neuroprotective, and cognition-enhancing substance with interesting potential in the treatment/prevention of Alzheimer’s disease. This disease is a progressive neurodegenerative disorder, characterized by the deposition of amyloid beta, neurofibrillary tangles, astrogliosis, and microgliosis, leading to neuronal dysfunction and loss in the brain. The pharmacological treatment for Alzheimer’s disease is only symptomatic, and focuses on cholinergic transmission. Apigenin could represent a novel tool to delay the onset of Alzheimer’s disease or slow down its progression [118].

The dietary accessibility of apigenin could represent a successful long-term treatment to prevent microglial activation and protect against or delay Alzheimer’s disease onset. Zhao et al. [92] and [93] tested the neuroprotective effects of apigenin in the amyloid precursor protein (APP/PS1) double transgenic Alzheimer’s disease mouse treated orally with 40 mg/kg of apigenin for three months. Improvements in memory and learning deficits as well as a reduction of fibrillar amyloid deposits with lowered insoluble concentrations of β-amyloid peptide, which is considered to play a critical role in the onset and progression of Alzheimer’s disease, were noted in the case of apigenin-treated mice. Additionally, it was shown that apigenin caused restoration of the ERK/CREB/BDNF pathway, involved in memory and typically affected in Alzheimer’s disease. Similarly, in another study, amnesia mouse models were treated with 20 mg/kg of apigenin. The results indicated improvements in spatial learning and memory, in addition to neurovascular protective effects [92]. Using a human induced pluripotent stem cell (iPSC)-derived model of Alzheimer’s disease, Balez et al. [119] reported that apigenin reduces neuronal hyper-excitability and apoptosis and inhibits the activation of cytokines and NO production, protecting Alzheimer’s disease neurons from inflammatory induced stress and neurite retraction.

Liang et al. [94] have investigated the therapeutic effect of apigenin on neuroinflammation in the glial fibrillary acidic protein-interleukin 6 (GFAPIL6)-expressing mouse using both immunohistochemical and behavioral tests. Histological staining showed that apigenin decreased the number of activated microglia of GFAP-IL6 mice both cerebellum and hippocampus by around 30% and 25%, respectively.

Popovic and colleagues [95] studied the effect of apigenin (20 mg/kg intraperitoneally (i.p.), 1 h before acquisition), on 24 h retention performance and forgetting of a step-through passive avoidance task, in young male Wistar rats. These workers reported that the pretreatment of apigenin caused a significant improvement in long-term memory but no significant effect on 24 h retention of fear memory.

Chamomile (*Matricaria chamomilla*) extract, considered a rich source of apigenin, was investigated as a natural agent for behavioral recovery (learning and memory functions) in scopolamine-induced dementia in a rat model [96]. Chamomile extract in doses of 200 mg/kg or 500 mg/kg of body weight per day, dosed for 15 days, showed significant memory-enhancing activity evaluated by the Morris water maze and passive avoidance paradigm models. The results obtained showed that *M. chamomilla* extract exhibited repairing effects on memory deficits induced by scopolamine, which was attributed to the free radical scavenging activity. They concluded that application of *M. chamomilla* ethanolic extract could have beneficial effects in the treatment of cognitive impairment of patients with Alzheimer’s disease and general behavioral disorders.

In another study, β-amyloid peptide-induced amnesia mouse models were treated with 20 mg/kg of apigenin [97]. Their results showed that apigenin application could improve spatial learning and memory, as well provide neurovascular protection. On the other hand, Zhang and co-workers [120] found that apigenin treatment reversed the decrease of superoxide dismutase and glutathione peroxidase activity, as well as increased the malondialdehyde level caused by spinal cord injury. The results suggested an antioxidant role of apigenin in response to the injury. What is more, decreased serum interleukin-1β (IL-1β), tumor necrosis factor-α (TNF-α) and intercellular adhesion molecule-1 release were observed, which suggested an anti-inflammatory effect of the tested polyphenolic compound. Only in a clinical study conducted by de Font-Reaulx Rojas and Dorazco-Barrag [102] did the authors achieve improvement in cognitive performance in humans with AD upon long-term administration of a formulation containing apigenin (every 12 h for 24 months).

#### 3.2.3. Apigenin’s Beneficial Effects in Depression and Insomnia

A study of the behavioral effects of acute administration of apigenin and chrysin, contained in *Matricaria chamomilla* and in *Passiflora incarnata*, has been conducted in rats. The results indicated that both compounds were able to reduce locomotor activity when injected in rats at a minimal effective dose of 25 mg/kg. The sedative effect could not be associated with an interaction with GABA–benzodiazepine receptors, since it was not counteracted by the benzodiazepine antagonist flumazenil [121]. Apigenin reduced GABA (gamma-aminobutyric acid)-activated Cl^−^ currents in a dose-dependent fashion. The effect was blocked by co-application of Ro 15-1788, a specific benzodiazepine receptor antagonist. Accordingly, apigenin reduced the latency in the onset of picrotoxin-induced convulsions. Moreover, apigenin injected intraperitoneally in rats reduced locomotor activity, but did not demonstrate anxiolytic or myorelaxant activity [122].

Weng et al. [98] studied the influence on depressive-like mice induced by chronic corticosterone treatment with 20 mg/kg or 40 mg/kg apigenin doses as well as fluoxetine (20 mg/kg). Behavioral tests showed that apigenin reversed the reduction of sucrose preference and the elevation of immobility time. In addition, the corticosterone-treated mice supplemented with apigenin ameliorated the decrease in hippocampal brain-derived neurotrophic factor (BDNF) levels underlining its antidepressant action by upregulation of BDNF. A randomized, long-term, clinical trial of the application of 500 mg three times per day of chamomile extract in the treatment of generalized anxiety disorder (GAD) was conducted by Mao and others [106]. For the whole study period, chamomile extract treatment was administered to eligible participants with a DSM-IV Axis-I diagnosis of GAD by 12 weeks of open label therapy. After that, 93 treatment responders were randomized in a double-blind, controlled trial to receive 26 weeks of either continuation of chamomile or placebo. Responders treated with chamomile, maintained significantly lower anxiety disorder symptoms than the placebo group. At the same time, chamomile participants showed reduced body weight and mean arterial blood pressure. Chamomile was safe and significantly reduced moderate-to-severe GAD symptoms. Amsterdam et al. [105] used chamomile extract in the treatment of GAD in a randomized, double-blind, placebo-controlled trial. Chamomile extract was standardized to a content of 1.2% apigenin. The volunteers had anxiety with co-morbid depression, or anxiety with past history of depression, or anxiety with no current or past depression. The results showed a significantly greater reduction in total Hamilton Depression Rating Scale (HAM-D) scores for chamomile, suggesting that *M. recutita* can exert an antidepressant effect. The unclear antidepressant action was attributed to chamomile’s flavonoids, possibly by modulation of noradrenalin (NA), dopamine (DA), and serotonin (5-HT) neurotransmission. Nakazawa et al. [99] found an antidepressant-like activity of apigenin on norepinephrine (NE) and dopamine (DA) turnover in the amygdala and hypothalamus in mice. In this regard, Han et al. [123] evaluated the effect of apigenin isolated from *Cayratia japonica* on MAO inhibition. Apigenin inhibited both MAO-A and MOA-B; the median inhibitory concentration (IC_50_) of MAO-A was 1.7 μM and for MAO-B it was 12.8 μM. Chaurasiya and colleagues [124] showed how inhibition of MAO-A by apigenin from propolis was 1.7-fold more selective than MAO-B. According to Lorenzo et al. [125], apigenin increases noradrenalin activity in an isolated rat atria model, at the same time inhibiting MAO activity in rat atria homogenates. On the other hand, Morita et al. [126] found that apigenin stimulated the uptake of L-tyrosine, a noradrenalin precursor. What is more, Yi and co-workers [100] in their work on antidepressant and neurochemical effects of citrus-associated apigenin found reduced immobility during the forced swim test (FST), reversed chronic mild stress (CMT)-induced reduction in sucrose intake in rats, lowered stress-induced alterations in 5-HT, DA, and reversed FST-induced increases in hypothalamic‒pituitary‒adrenocortical axis activity. What is more, it is believed that inflammation may contribute to the pathophysiology of depression.

In the study conducted by Li and others [101], the effects of apigenin on lipopolysaccharide (LPS)-induced depressive-like behavior in mice model were examined. Tested animals were pre-treated with 25 mg/kg or 50 mg/kg of apigenin or 20 mg/kg of fluoxetine once daily for seven days. The use of apigenin prevented the abnormal behavior induced by LPS, attenuated production of pro-inflammatory cytokines interleukin-1β (IL-1β) and tumor necrosis factor-α (TNF-α). What is more, the authors found that apigenin suppressed inducible nitric oxide synthase (iNOS) and cyclooxygenase-2 (COX-2) expression. Apigenin, at a dose of 50 mg/kg, reversed the depressive-like behavior induced by tumor necrosis factor-α without altering the locomotor activity. They concluded that, due to its anti-inflammatory properties, apigenin is characterized by antidepressant-like properties in LPS-treated mice.

Insomnia is a prevalent sleep disorder that can profoundly impact a person’s health and well-being [127]. Chamomile flower extract, with more than 2.5 mg of apigenin, was examined for its preliminary efficacy and safety for improving sleep and daytime symptoms in patients with chronic insomnia. Thirty-four adults aged 18–65 years with primary insomnia (DSM-IV criteria) lasting more than six months, with total daily sleep time less than 6.5 h, took part in research carried out by Zick et al. [103]. They found no significant differences between groups in changes in sleep diary measures, including total sleep time, sleep efficiency, sleep latency, wake after sleep onset, sleep quality, and number of awakenings. It should be highlighted that chamomile did cause a modest improvement in daytime functioning. The authors concluded that chamomile treatment could provide modest benefits in terms of daytime functioning and mixed benefits in terms of sleep diary measures.

Due to the fact that both insomnia and depression are central nervous system (CNS)-related diseases, the factor affecting the effectiveness of bioactive compounds is blood‒brain barrier (BBB) penetration. The role of the BBB is to provide nutrients for the brain as well as regulate the brain microenvironment for neuronal functions [128]. Consequently, BBB protects the central nervous system from compounds that can negatively affect the function of the CNS. Some studies have demonstrated that flavonoids can easily penetrate the BBB [129]. According to Yang et al. [130], the permeation order of the flavonoids is genistein > isoliquiritigenin > apigenin > puercetin > kaempferol > hesperidin > rutin > quercetin, where genistein is characterized by the highest permeation [130]. Thus, apigenin may have direct, positive effects on diseases such as AD or insomnia.

#### 3.2.4. Anticancer Effects of Apigenin

In general, due to the source of apigenin, it appears as one of the bioactive compounds of plant origin, which reduces the incidence of cancer. It is known that high intake of flavonoids from vegetables and fruits can be inversely associated with the risk of cancer. Knekt et al. [131] investigated the association between flavonoid (quercetin, kaempferol, myricetin, luteolin, and apigenin) intake and lung cancer. They found an inverse association between the intake of flavonoids and incidence of all sites of cancer, which also provides strong evidence of a protective role of flavonoids against lung cancer. The authors concluded that apples as well as onions, which are a source of apigenin, show a protective role against lung cancer. The association between dietary flavonoids and their protective role as well as reduction of risk of cancer was investigated, among others, in the studies conducted on ovarian cancer [132], breast cancer [133], and the recurrence risk of neoplasia with resected colorectal cancer patients [134].

Madunić et al. [11] reported the possible chemotherapeutic modality of apigenin due to its low intrinsic toxicity and remarkable effects on normal versus cancerous cells, compared with other structurally related flavonoids. The authors reviewed the apigenin anticancer activities including dose ranges used in both in vitro and in vivo studies. Silvan et al. [85] investigated the chemopreventive potential of apigenin during 7,12-dimethylbenz(a)anthracene (DMBA)-induced hamster buccal pouch carcinogenesis. Apigenin was simultaneously given at a dose of 2.5 mg/kg body weight/day, starting one week before exposure to the carcinogen and continuing to the end of the experiment. The obtained results showed that apigenin, in comparison to the control sample (DMBA only), prevented tumor formation. Although mild to moderate pre-neoplastic lesions (hyperplasia, hyperkeratosis, and dysplasia) were observed in this group of hamsters, the authors found that oral administration of apigenin also brought back the status of lipid peroxidation, antioxidants, and phase I and phase II detoxification agents to near normal range during DMBA-induced oral carcinogenesis.

The therapeutic antitumor effects of apigenin were evaluated using an in vivo mouse model by Chuang et al. [86]. For the treatment of the E7-expressing tumor (TC-1), the authors used apigenin only and a combination of apigenin with DNA vaccines encoding the HPV-16 E7 antigen linked to heat shock protein 70 (HSP70). Apigenin (25 mg/kg) was implemented three days after TC-1 implantation and continued for 10 days. In the case of the E7-hsp70-only group, three days after TC-1 implantation, each mouse was first vaccinated with 2 µg of E7-hsp70 via a gene gun, and boosted seven days later. Finally, in the case of the combination of apigenin with E7-hsp70, after TC-1 implantation, each mouse received the same vaccination schedule as the E7-hsp70 and the same apigenin schedule as the apigenin group. It was found that treatment with apigenin rendered the TC-1 tumor cells more susceptible to lysis by E7-specific cytotoxic CD8+T cells and enhanced apoptotic tumor cell death. The results showed that mice treated with apigenin combined with E7-HSP70 DNA had the highest frequency of primary and memory E7-specific CD8+T cells, leading to potent therapeutic anti-tumor effects. They concluded that apigenin represents a promising chemotherapeutic agent, which may be used in combination with immunotherapy for the treatment of cancers [86].

Quercetin, apigenin, epigallocatechin gallate (EGCG), resveratrol, curcumin, and tamoxifen were investigated as inhibitory agents of the growth and metastatic potential of B16-BL6 melanoma cells in vivo in a syngeneic mouse model [87]. They found that quercetin at 25 and 50 mg/kg as well as apigenin at 25 and 50 mg/kg, and epigallocatechin gallate, resveratrol, and tamoxifen at 50 mg/kg, significantly delayed tumor growth. They found that at the dose of 50 mg/kg, EGCG, apigenin, and quercetin were the most effective. In addition, cisplatin, at 2 mg/kg, significantly inhibited melanoma growth, and apigenin, at 25 mg/kg, potentiated the effect without mortality or body weight loss. Furthermore, the authors found that quercetin and apigenin, at 25 mg/kg as well as 50 mg/kg, significantly decreased the number of lung metastatic colonies, while other tested polyphenols were not active even at higher concentrations. Thus, it was concluded that apigenin not only inhibits the growth of melanoma cells, but also shows anti-invasive potential.

Torkin and co-workers [88] investigated the effect of apigenin in doses of 25 mg/kg as an inhibitory agent of neuroblastoma tumor in mice. They found that mice bearing scapular NUB-7 tumors, treated with apigenin for five days, did not display overt toxicity relative to untreated mice. The survival of primary sympathetic neurons was not inhibited, thus apigenin was not toxic to non-transformed cells. At the same time, tumor mass in the treated group of mice decreased by 50%. They indicated that the inhibition of NUB-7 xenograft tumor growth in a non-obese diabetic/severe combined immunodeficiency mouse model was likely caused by inducing apoptosis. The authors presumed that the mechanism of action of apigenin involved p53, as it increased the levels of p53 and the p53-induced gene products. They concluded that apigenin can be a candidate for neuroblastoma treatment that likely acts by regulating a p53-Bax-caspase-3 apoptotic pathway.

Apigenin activity was a subject of investigation in the treatment of prostate cancer conducted by Shukla and others [89]. Transgenic adenocarcinoma mouse prostate (TRAMP) models were treated with 20 μg/day or 50 μg/day of apigenin for six days per week for 20 weeks. The authors observed significant volume reduction of prostate tumors as well as completely abolishment of distant organ metastasis. According to their results, apigenin caused significant decrease in the weight of genitourinary apparatus, dorsolateral as well as ventral prostate. Treated mice showed reduced phosphorylation of Akt and FoxO3a transcription factor, and the results correlated with increased nuclear retention and decreased binding of FoxO3a with 14-3-3 protein. In general, Akt, which phosphorylates FoxO3a at multiple sites, facilitating its association with 14-3-3, negatively regulates FoxO3a activity and as a consequence, leads to its transport out of nucleus to cytoplasm. The results obtained by Shukla and colleagues [89] provide evidence that apigenin can effectively suppress prostate cancer progression, at least in part by targeting the PI3K/Akt/FoxO-signaling pathway. On the other hand, in two previous studies by Shukla et al. [90,91], the authors found that apigenin suppresses the in vivo growth of prostate cancer by targeting β-catenin and insulin-like growth factor-I-signaling pathways.

#### 3.2.5. Other Effects of Apigenin

Apigenin has been considered as a potential natural treatment for inflammatory disorders of the central nervous system, such as multiple sclerosis. There is, however, a gap in information about the molecular mechanism of action of apigenin leading to its modulatory effects on dendritic cells responsible for maintaining immune balance. It has been recently shown that apigenin can reduce cytoplasmic RelB levels in LPS-treated DCs isolated from normal peripheral blood of a healthy donor. A decrease in glucose uptake and glycolysis has also been observed, together with an increase in mitochondrial activity [135]. Another work of Kim et al. [136] evaluated the anti-inflammatory properties of an apigenin di-*C*-glycosides present in natural extracts of *Camellia, Viscum*, and *Korthalsella japonica* on a murine monocyte/macrophage cell line (RAW 264.7).

Apigenin has been associated with antiviral effects, together with quercetin, rutin, and other flavonoids. The antiviral activity appears to be connected to the non-glycosidic compounds, and hydroxylation at the 3-position is apparently a prerequisite for antiviral activity. Apigenin has also been reported to exhibit anti-inflammatory activity [137]. Nielsen et al. [138] carried out a two-week randomized crossover trial studying the effect of intake of parsley, *Petroselinum crispum*, containing high levels of apigenin, on the urinary excretion of flavones and on biomarkers for oxidative stress. The study showed that, in a diet supplemented using parsley containing 3.73 to 4.49 mg apigenin/MJ in 24 h, the fraction of apigenin intake excreted in the urine was 0.58%. Erythrocyte glutathione reductase and superoxide dismutase activities increased during intervention with parsley as compared with the levels on the basic diet [138].

Shoara et al. [104] conducted a randomized controlled clinical trial on the efficacy and safety of apigenin-rich chamomile oil for knee osteoarthritis. A treatment with topical chamomile oil three times/day for three weeks significantly reduced the demand of analgesic (acetaminophen) of patients with knee osteoarthritis, improving their physical function.

Sui and co-workers [139] tested the apigenin effect on the expression of angiotensin-converting enzyme 2 (ACE2) in the kidneys of spontaneously hypertensive rats. They found that the transcription level of angiotensin-converting enzyme 2 mRNA in the positive control group and the group of rats treated with 0.417 g/kg of apigenin was significantly higher than in the control group. The authors found that the lower blood pressure effect of apigenin is given by upregulating the expression of ACE2 in kidney.

Tamayose et al. [140] studied the antioxidant properties of apigenin derivative (8-*C*-rhamnosyl apigenin (8CR)) from *Peperomia obtusifolia* in significant mitigation of the pharmacological effect induced by sPLA2 from snake venom. A preventive (15 min) intraperitoneal injection of 8CR at a concentration adjusted to 200 μg (8 mg/kg) in rat significantly reduced edema and the myotoxic effect induced by sPLA2.

### 3.3. Limitations and Future Perspectives

Apigenin is considered safe, even at high doses, and no toxicity has been reported. Nonetheless, at high doses, it can trigger muscle relaxation and sedation [141]. The direction towards micro- and nanodelivery for therapeutic formulations containing apigenin should be evaluated and carried out, both to better target tissues and organs and to enhance the therapeutic efficacy. The possibility of modulating the delivery of apigenin through a controlled release of this active compound should be explored [142]. This aspect is a challenge in food-derived bioactive compounds, which can act as nutraceuticals and/or supplements. Thanks to the widely accepted benefits of apigenin, there are possibilities emerging to establish new methods for the recovery of this compound from alternative sources, such as macro- and microalgae [143,144] and agro-food waste [145], achieved through using green and innovative technologies, like enzymatic treatment, microwave-assisted extraction, ultrasound-assisted extraction, supercritical CO_2_ and subcritical water technologies, which can be both environmentally friendly and sustainable [146,147].

## 4. Apigenin in Databases

Generally, studies that examine the relationship between diet and health have led to increased interest in all biologically active constituents present together with nutrients in food, and data on these, as well as other compounds, are increasingly required in the database system; detailed and structured information of bioactive compounds in foods, leading to complete and comprehensive harmonized databases on the content, is crucial in dietary assessment and in the evaluation of a formulated diet to investigate health effects in vivo [148]. The implementation of databases has emerged, based on both analytical data and data taken from the literature, throughout a harmonized and standardized approach for the evaluation of an adequate dietary intake [149].

The main public databases, gathering extensive data on the flavonoids, including apigenin, content of foods, and beverages, are: the United States Department of Agriculture (USDA) [150,151], Phenol-Explorer [152,153], and eBASIS (Bioactive Substances in Food Information Systems) [154,155,156].

Phenol-Explorer comprise the first comprehensive web-based open-access database on the content of polyphenol in foods; further updates include data on pharmacokinetic and metabolites, the effect of food processing, and cooking [157,158]. The data were collected from peer-reviewed scientific publications, and evaluated before they were aggregated to produce final representative mean content values. To date, the Phenol-Explorer data on apigenin were present for apigenin and derived forms (i.e., apigenin 6-*C*-glucoside, apigenin 7-*O*-glucuronide, apigenin 6,8-*C*-arabinoside-*C*-glucoside, apigenin 7-*O*-(6″-malonyl-apiosyl-glucoside). For apigenin, the data encompass the following food groups and subgroups: alcoholic beverages, fruit and vegetable oils, herbs, nuts, and vegetables (i.e., leaf vegetables, fruit vegetables, root vegetables, onion-family vegetables). For example, within the fruit and vegetable oils, for extra virgin olive oil, a mean content of 1.17 mg/100 g FW was reported (produced from 17 original content values extracted from three different publications); for refined olive oil, 0.03 mg/100 g FW (produced from three original content values extracted from one publication); for virgin olive oil, 0.10 mg/100 g FW (produced from 69 original content values extracted from six publications) [152].

The USDA database was developed in 2004 and has included flavonoids in subsequent versions [151], based on a compilation of data from the literature. The USDA update on reported apigenin data in food items was subdivided into: dairy and egg products, spices and herbs, fats and oils, soups, sauces and gravies, fruits and fruit juices, vegetables and vegetable products, nuts and seeds, beverages, baked products, and sweets [151].

eBASIS represents the first EU harmonized food composition database, containing composition data and biological effects of over 300 major European plant foods of 24 compound classes in 15 EU languages [156]. It is based on the compilation of critically evaluated data from peer-reviewed literature, inserted as raw data. Currently, there are 1039 composition data points in eBASIS for flavones, including data for apigenin as its aglycone and as apigenin glycosides [154,156]. Data for beneficial effects cover in vitro, in cell, and in vivo studies.

It should be underlined, at the same time, that the understanding of activities of bioactive compounds in humans is a key issue; the work by Dragsted et al. [159] marked the importance of databases for dietary and health biomarkers. There are three databases related to the role of biologically active compounds and their metabolites in humans: the Human Metabolome Database or HMDB 4.0 [160], a web metabolomic database on human metabolites [161], and PhytoHub [162,163].

## 5. Conclusions

Based on the in vitro and in vivo evidence reported here, apigenin, a natural bioactive flavone-type molecule, could play a key role in the prevention and treatment of emerging global health issues, highlighting once again the significant use of food components and/or plant compounds. Overall, it is very difficult to get general or unequivocal information regarding its prophylactic function within the human body, its bioavailability and bioactivity due to high inter-individual variability, and the several mechanisms of its biological actions affecting human health. The possibility that different components present in the supplemented mixtures can interact, generating antagonistic, synergistic, or additive effects, makes it difficult to predict the function, or to differentiate between prevention and therapy. The use of apigenin among patients under conventional pharmacological therapy should be carefully addressed, given the high likelihood of food‒drug interactions. There is a need to establish new and adequate cellular and animal models, which may, in turn, allow the design of more efficient and prevention-targeted clinical studies using apigenin and/or its derivatives as candidates for therapeutic drugs in the near future.

## Figures and Tables

**Figure 1 ijms-20-01305-f001:**
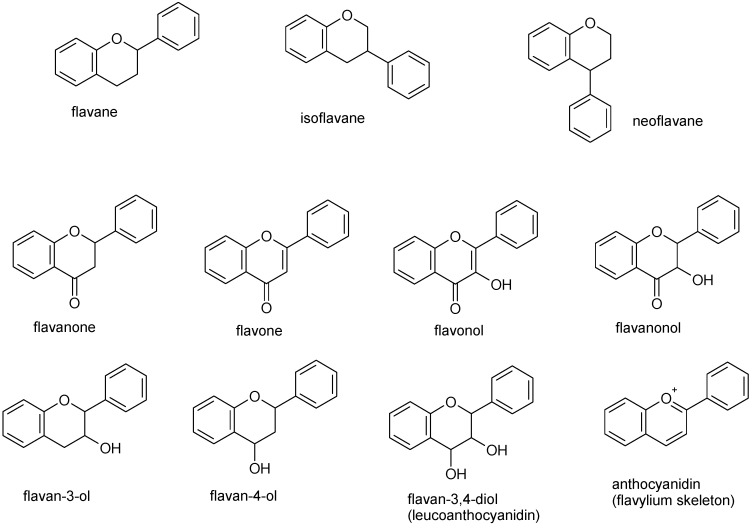
Basic structure of the flavane, isoflavane, and neoflavane backbones, flavanone (2,3-dihydro-2-phenylchroman-4-one) and of the various classes of flavonoids.

**Figure 2 ijms-20-01305-f002:**
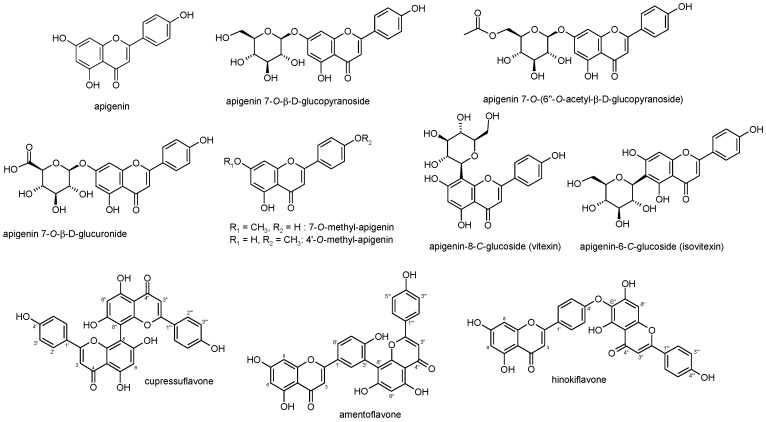
Structures of apigenin and its glycosidic, glucuronide, acetylated, and methyl ester derivatives together with some biflavonoids of apigenin.

**Figure 3 ijms-20-01305-f003:**
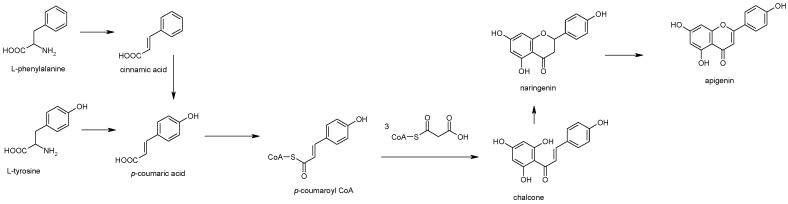
Biogenetic pathway of apigenin biosynthesis.

**Figure 4 ijms-20-01305-f004:**
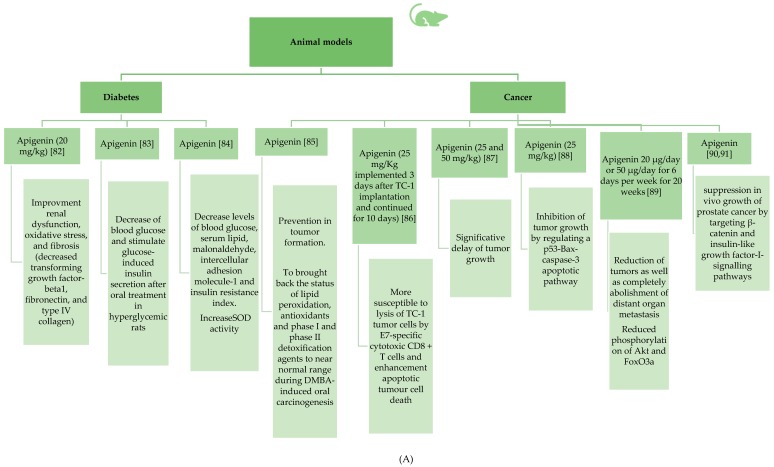
(**A**) Animal model studies involving apigenin effects in diabetes and cancer [82,83,84,85,86,87,88,89,90,91]. (**B**) Animal model studies involving apigenin effects in Alzheimer’ disease and amnesia [92,93,94,95,96,97]. (**C**). Animal model studies involving apigenin effects in depression [98,99,100,101].

**Figure 5 ijms-20-01305-f005:**
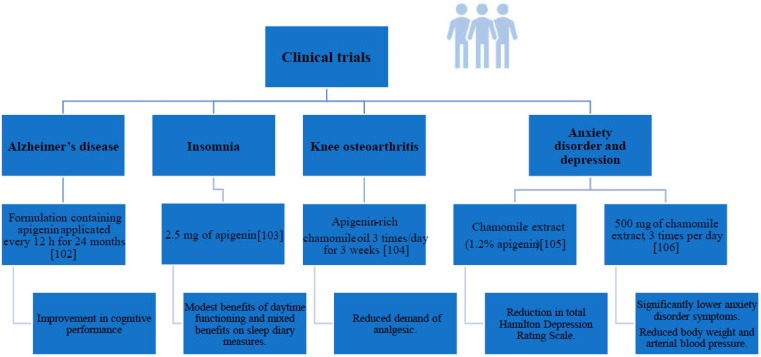
Human studies monitoring apigenin supplementation [102,103,104,105,106].

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
