# Peer review of "The Therapeutic Potential of Apigenin"

_ijms, 2019, doi:10.3390/ijms20061305_

Reviewer 1 Report

This is a comprehensive review made by Salehi et al. to describe the therapeutic potentials of apigenin in various aspects, including cancer therapy, antidiabetic properties, Alzheimer’s disease, depression and so on. The review mainly focuses on animal experiments and clinical trials in an appropriate format. I have only two minor opinions

     1. The studies done with plant extracts is hard to demonstrate that apigenin has a critical role in those functions. Especially in the study regarding the effectiveness of Ocimum sanctum (ref. 134). Moreover, even in that paper, the authors did not mention the role of apigenin in the article. I recommend that this article is not suitable to be included in this review.

2. In table 2, the words “clinical trails” should be “clinical trials”.

Author Response

The authors want to express their gratitude to the reviewer for his/her valuable comments and suggestions. The authors’ replies to the individual points raised are reported in Italic below.

Open Review

(x) I would not like to sign my review report

( ) I would like to sign my review report

English language and style

( ) Extensive editing of English language and style required

 ( ) Moderate English changes required

(x) English language and style are fine/minor spell check required

( ) I don't feel qualified to judge about the English language and style

This is a comprehensive review made by Salehi et al. to describe the therapeutic potentials of apigenin in various aspects, including cancer therapy, antidiabetic properties, Alzheimer’s disease, depression and so on. The review mainly focuses on animal experiments and clinical trials in an appropriate format. I have only two minor opinions

1. The studies done with plant extracts is hard to demonstrate that apigenin has a critical role in those functions. Especially in the study regarding the effectiveness of Ocimum sanctum (ref. 134). Moreover, even in that paper, the authors did not mention the role of apigenin in the article. I recommend that this article is not suitable to be included in this review.

As suggested, this reference was not included in this review and  the part of the text related to the reference was deleted.

2. In table 2, the words “clinical trails” should be “clinical trials”.

The words “clinical trails” was replaced by “clinical trials”

Reviewer 2 Report

In this manuscript the authors summarize the health promoting effects of apigenin focusing on diabetes, amnesia, Alzheimer’s disease, depression, Insomnia and Cancer in particular through in vivo research.  In general, this is an interesting topic, however, some modification are necessary.  For example, more discussion, authors point of the view, and new section (e.g. limitation, risk of Apigenin and future perspectives.) should be included.

In general more discussion as well as authors point of      the view in each section should be included.

Apigenin as nutraceutical andHealing properties section are overlapping. I suggest  include absorption, metabolism, distribution and excretio in nutraceutical section and  medical/therapeutical properties of AP in other section.

Why only few studies have been included when authors summarized the main results      reviewed in animals?. For example in AD there are several in vivo studies available      https://doi.org/10.1016/j.phrs.2017.10.008. I suggest design a new table and include more AD and depression studies.

A new figure summarizing the mechanisms of AP in each pathology included in the review will help to easy catch the main concepts.

Apigenin in database section is weak, should be improved. 

Additional section LIMITATIONS AND FUTURE PERSPECTIVE should be included.

Only one figure summarizing the main structures of some derived flavonoids of API and API biosynthesis should be included.

Recently,  the antioxidative activity and potential      role as neuroprotective agent of Apigenin, its chemistry, pharmacokinetic and metabolism in the context of depression, Ad and PD has been summarized https://doi.org/10.1016/j.phrs.2017.10.008. However, this manuscript has not been discussed by the authors.

English can be revised throughout the text by fixing grammatical mistake and reducing the      sentence length.

Author Response

The authors want to express their gratitude to the reviewer for his/her valuable comments and suggestions. The authors’ replies to the individual points raised are reported in Italic below.

Review Report Form

In this manuscript the authors summarize the health promoting effects of apigenin focusing on diabetes, amnesia, Alzheimer’s disease, depression, Insomnia and Cancer in particular through in vivo research. In general, this is an interesting topic, however, some modification are necessary. For example, more discussion, authors point of the view, and new section (e.g. limitation, risk of Apigenin and future perspectives.) should be included. In general more discussion as well as authors point of the view in each section should be included.

The discussion was implemented following your suggestions.

Apigenin as nutraceutical and Healing properties section are overlapping. I suggest  include absorption, metabolism, distribution and excretion in nutraceutical section and  medical/therapeutical properties of AP      in other section.

As suggestied we have moved absorption, metabolism, distribution and excretion in nutraceutical section and medical/therapeutical properties of apigenin   in healing properties section or further paragraphs.

Why only few studies have been included when authors summarized the main results reviewed in animals? For example in AD there are several in vivo studies available https://doi.org/10.1016/j.phrs.2017.10.008. I suggest design a new table and include more AD and depression studies. A new figure summarizing the mechanisms of AP in each pathology included in the review will help to      easy catch the main concepts.

The Figure 4 was implemented, by inserting proper studies that you suggested, and was structured in Figures 4A, 4B and 4C and implemented with more details. Studies related to each pathology are reported in Figures 4A, 4B, 4C and Figure 5.

Apigenin in database section is weak, should be improved.

This section was improved.

Additional section LIMITATIONS AND FUTURE PERSPECTIVE should be included.

As suggested, additional section was included.

Only one figure summarizing the main structures of some derived flavonoids of API and API biosynthesis should be included.

The number of figures summarizing the main structures were reduced.

Recently, the antioxidative activity and potential role as neuroprotective agent of Apigenin, its chemistry, pharmacokinetic and metabolism in the context of depression, Ad and PD has been summarized https://doi.org/10.1016/j.phrs.2017.10.008. However, this manuscript has not been discussed by the authors.

The proper reference that you suggested is inserted in order to better describe the antioxidative activity and potential role as neuroprotective agent of Apigenin .

English can be revised throughout the text by fixing grammatical mistake and reducing the sentence length.

The manuscript has been reviewed by a native speaker of English to improve the English grammatical mistakes.

Round  2

Reviewer 2 Report

Still the manuscript need minor English language editing and spell check. 

All references should be verified, some authors last mane are wrong in the text. For example, line 267. The recent review of Nasabi et al. [117], however reference 117 is not Nasabi.

Author Response

The authors want to express their gratitude to the reviewer for his/her valuable comments and suggestions. The authors’ replies to the individual points raised are reported in Italic below.

Still the manuscript need minor English language editing and spell check. 

The manuscript has been reviewed by a native speaker of English.

All references should be verified, some authors last mane are wrong in the text. For example, line 267. The recent review of Nasabi et al. [117], however reference 117 is not Nasabi.

In the text the reference of Nabavi et al. 2018 was correctly reported in the text. Moreover , all references were checked.